# Gear Fault Diagnosis Based on Kurtosis Criterion VMD and SOM Neural Network

**Dongming Xiao [1,2], Jiakai Ding [1,2] , Xuejun Li [1,*] and Liangpei Huang [2]**

[1]  School of Mechatronics Engineering, Foshan University, Foshan 528225, China; dongming.xiao@outlook.com (D.X.); jiakai1996@outlook.com (J.D.)

[2]  Hunan Provincial Key Laboratory of Health Maintenance for Mechanical Equipment, Hunan University of Science and Technology, Xiangtan 411201, China; huanglp413@163.com

[*]  Correspondence: hnkjdxlxj@163.com; Tel.: +86-135-0732-2252

**Abstract:** A gear fault diagnosis method based on kurtosis criterion variational mode decomposition (VMD) and self-organizing map (SOM) neural network is proposed. Firstly, the VMD algorithm is used to decompose the gear vibration signal, and the instantaneous frequency mean is calculated as the evaluation index, and the characteristic curve is drawn to screen out the most relevant intrinsic mode functions (IMFs) of the original vibration signal. Then, the number of VMD decompositions is determined, and the kurtosis value of IMFs are extracted to form the feature vectors. Then, the kurtosis value feature vectors of IMFs are normalized to form the kurtosis value normalized vectors. Finally, the normalized vectors of kurtosis value are input into SOM neural network to realize gear fault diagnosis. When the number of training times of SOM neural network is 100, the gear fault category is accurately classified by SOM neural network. The results show that when the training times of SOM neural network is 100 times, the gear fault diagnosis method, based on the kurtosis criterion VMD and SOM neural network is 100%, which indicates that the new method has a good effect on gear fault diagnosis.

**Keywords:** variational mode decomposition; kurtosis value; SOM neural network; instantaneous frequency mean; gear fault diagnosis

## 1. Introduction

The gearbox is an important part of the mechanical transmission. The failure of the gear in the gearbox will lead to the damage of the system, which will be paralyzed in serious cases. Therefore, gear fault monitoring and diagnosis is an important means to prevent system damage. In the operation of the gearbox, the vibration signal generated by the gear failure is weak and difficult to monitor. Therefore, it is especially important to monitor the vibration signal of gear faults.

Vibration signal monitoring is an important means of gear fault diagnosis [1]. In the past, the sensor signal transmission in industrial monitoring was transmitted by wire, but with the rise of the Internet of things (IoT), it has become an important technology in the monitoring system [2]. For offshore wind turbines, the gear is usually used for wind turbines, but due to the limitation of its internal structure, there is not enough space for sensors to monitor signals for fault diagnosis, so an inductive thermal imaging method is proposed to detect gear faults by Gao [3]. In the aspect of gear fault monitoring based on data, a signal processing method is used to process gear fault signal. Traditional signal processing methods have been widely applied to gear fault diagnosis, such as time-frequency analysis [4], wavelet analysis [5,6], Hibert-Huang transform (HHT), empirical mode decomposition (EMD) [7], and local mode decomposition (LMD) [8]. However, the wavelet transform is based on the analysis of the specified wavelet base. Once the specified wavelet base is specified,

the decomposed mode is fixed. It is better to use different wavelet bases for the analysis of different signals to achieve the best processing effect. EMD and LMD are prone to endpoint effect and mode mixing during the decomposition process. Therefore, Dragomiretskiy [9] proposed a variational mode decomposition to solve the problems caused by EMD and LMD in the decomposition process. Wang [10] used variational mode decomposition (VMD) to extract various fault features of the gearbox under strong noise environment, and compared with the ensemble empirical mode decomposition (EEMD) decomposition results, it shows that the algorithm can effectively improve the signal-to-noise ratio of the signal. Li [11] proposed a fault diagnosis method based on VMD and generalized composite multi-scale dynamic entropy (GCMSDE) to identify different health conditions of planetary gearboxes. Feng [12] uses VMD to decompose the planetary gearbox vibration signal into several intrinsic mode functions (IMFs), and performs Fourier transform on the amplitude envelope and instantaneous frequency of the sensitive IMFs to obtain the amplitude and frequency demodulation spectrum. The planetary gearbox faults have been detected based on demodulation and have been successfully identified on all three gears (sun gear, planetary gear, and ring gear). Wang [13] used the improved VMD algorithm to diagnose the gearbox and compared it with EEMD to verify the effectiveness of the proposed method. Si [14] proposes an improved VMD linked wavelet denoising method, which can suppress high frequency narrowband noise and normal noise in electromagnetic acoustic transducer (EMAT) signal, and this method can retain defect information.

In recent years, researchers have studied a large number of fault classification algorithms. Among them, the gear fault classification algorithms mainly include support vector machine (SVM) [15], artificial neural network (ANN) [16], and deep learning [17]. Chen [18] proposed a gearbox fault diagnosis model based on wavelet support vector machine. The results show that it has stronger generalization ability and higher diagnostic accuracy than artificial neural network and support vector machine with random extraction parameters. Bordoloi [19] used different optimization methods to optimize SVM parameters and used continuous wavelet transform (CWT) and wavelet packet transform (WPT) for feature extraction. The results showed that when time-domain signals were used, their classification ability was lower than their prediction ability. Compared with SVM, neural network has strong generalization ability, so it is widely used in fault diagnosis. Kohonen [20] proposed a self-organizing map (SOM) neural network, which utilizes various features in the signal and the characteristics of the internal representation of its spatial organization for self-organization and self-learning, and then hierarchical clustering. A gear fault diagnosis method based on HHTand SOM neural network is proposed by Cheng [21], which used EMD to decompose gear vibration signals to obtain several IMFs, and selected the energy percentage of the first six IMFs as the input vector of SOM neural network for fault identification. The analysis results show that this method can effectively identify gear fault types. The SOM-radial basis function (RBF) neural network to detect and analyze the fault of induction motor is used by Wu [22]. The results show that the method can not only detect electrical and mechanical faults, but also estimate the degree of fault.

Based on the studies in the above literatures, due to the problems of mode mixing and endpoint effect in EMD and LMD methods, VMD was adopted for signal decomposition in this paper.At the same time, VMD is difficult to determine the penalty factor and the number of modal decomposition in the decomposition process, so the instantaneous frequency mean change of each component of the fault signal after VMD decomposition is put forward as an evaluation index, so as to determine the number of modal decomposition in the VMD process. At the same time, in order to highlight the characteristics of the signal, the kurtosis value is used as the characteristic parameter of the signal to extract, and a gear fault diagnosis based on the kurtosis criterion VMD and SOM neural network is proposed.

## 2. Experiment and the Procedure

### 2.1. Experimental System and Data Acquisition

In order to verify the effectiveness of the proposed kurtosis criterion VMD and SOM neural network method in gear fault diagnosis, a spiral bevel gear test rig was established and tested. The experimental system is shown in Figure 1. The test rig includes a three-phase asynchronous motor (7) for driving, a speed reducer (5), and an electromagnetic speed control motor controller (8) for motor speed regulation. The reducer (5) is connected to the output shaft by a coupling (3), and the speed of the three-phase asynchronous motor (7) is controlled by a speed controller, which can operate the reducer (5) to be tested at various speeds. The reducer (5) to be diagnosed is Shanghai Nini reducer, the model is T6-3/1-LR (Shanghai Nini reducer Co., Ltd, Shanghai, China), which is driven by three-phase asynchronous motor (7) through coupling (6). The rated power output of three-phase asynchronous motor (7) (Shanghai Bengxin electric motor Co., Ltd, Shanghai, China) is 1.1kW. Among them, the electromagnetic speed-regulating motor controller (8) (Shanghai Shanchuang instrument & meter Co., Ltd, Shanghai, China) allows manual adjustment of load torque. The vibration signal of the reducer (5) is collected by the acceleration sensor (4), which adopts 4514b-001 acceleration sensor (4) produced by B&K company in Denmark. Among them, the acceleration sensor (4) output is sent to laptop (1) through data acquisition analyzer (B&K, Type 3053-b120, B&K company, Denmark) (2). The acceleration sensor (4) layout position is at the output shaft of gear reducer (5). The B&K data acquisition analyzer (2) is used to collect the vibration signals of the gears when it is, so as to simulate the deterioration process of gear fault. The failure parts of normal gear, gear with tooth wear, gear with tooth crack, and gear with tooth break in the spiral bevel gear test rig is shown in Figure 2.

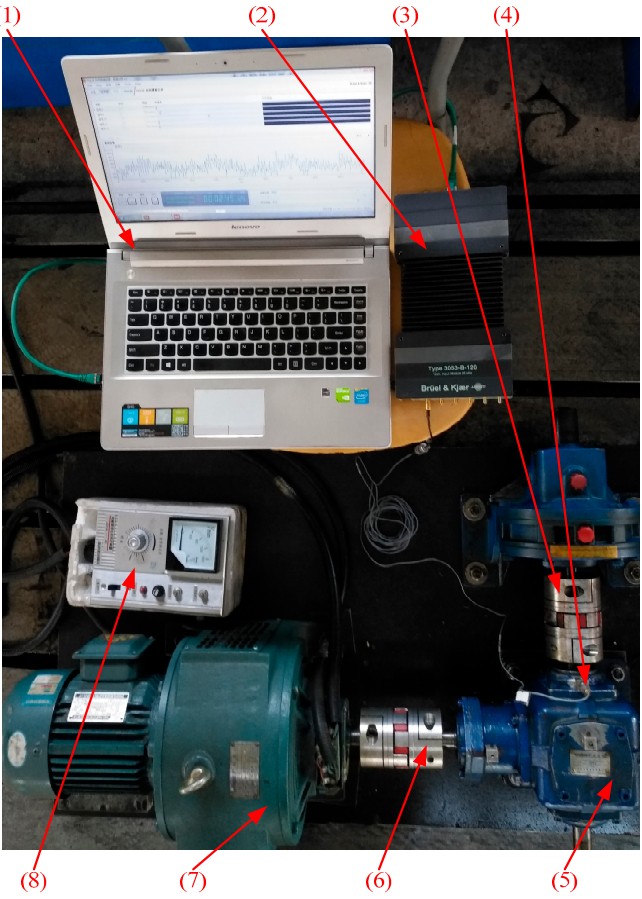

**Figure 1.** Experimental system: (1) laptop, (2) B&K data acquisition analyzer, (3) coupling, (4) acceleration sensor, (5) reducer, (6) coupling, (7) three-phase induction motor, and (8) electromagnetic speed control motor controller.

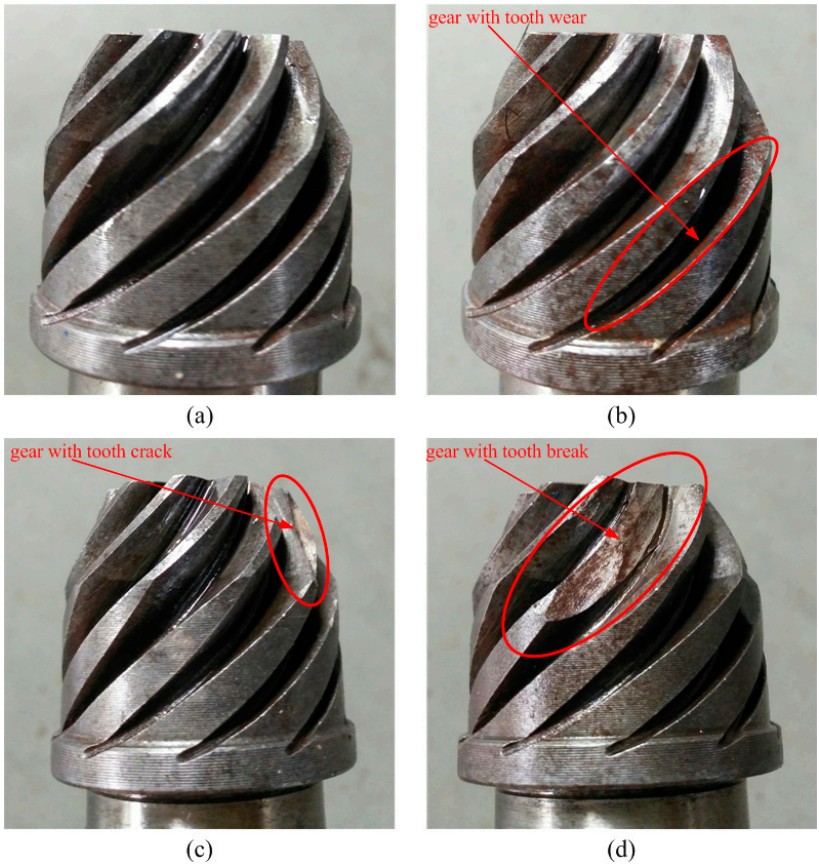

**Figure 2.** (**a**) Normal gear, (**b**) gear with tooth wear, (**c**) gear with tooth crack, and (**d**) gear with tooth break.

In this paper, normal gear, gear with tooth wear, gear with tooth crack, and gear with tooth break under four conditions were analyzed. Among them, the sampling frequency of gear vibration signal is 8192Hz, the acquisition duration of each signal is 0.25s, and a total of 16 segments of data are collected.

### 2.2. Fault Diagnosis Process Based on Kurtosis Criterion VMD and SOM Neural Network

Noise will inevitably be introduced in the process of gear vibration signal acquisition, which will cause adverse impact on fault diagnosis. Therefore, this paper proposes a method based on kurtosis criterion VMD and SOM neural network for gear fault diagnosis and identification. Fault diagnosis flow chart is shown in Figure 3:

The specific steps are as follows:

Step 1. According to a certain sampling frequency $f_s$, vibration signals of gear under four working conditions of normal gear, gear with tooth wear, gear with tooth crack and gear with tooth break for $N$ times were collected, with a total of $4N$ samples;

Step 2. VMD decomposition was carried out for each vibration signal, and a total of $k$ IMFs were decomposed;

Step 3. Extract the kurtosis value of each IMFs to form the feature vectors;

Step 4. Normalized feature vectors;

Step 5. Input normalized vector to SOM neural network for fault diagnosis;

Step 6. Output fault diagnosis results.

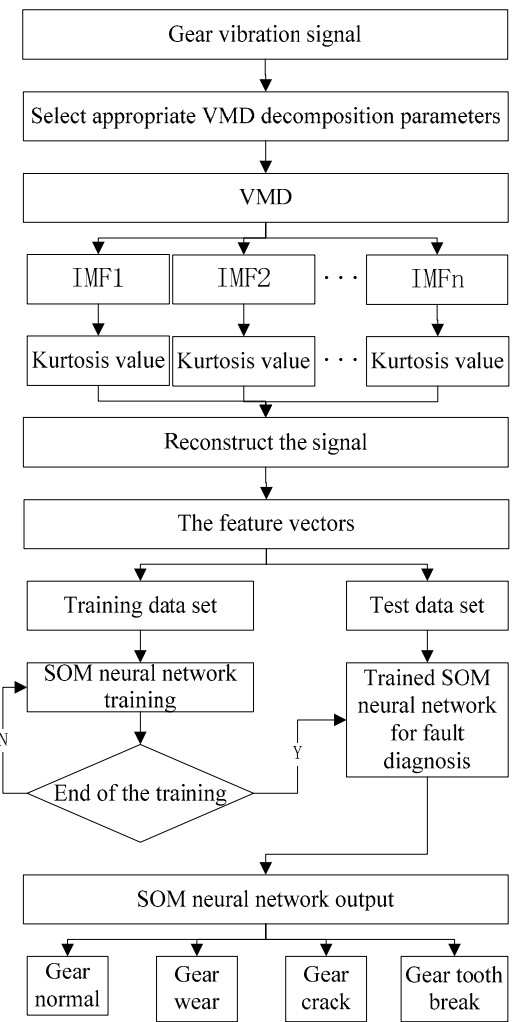

**Figure 3.** Fault diagnosis flow chart based on kurtosis criterion variational mode decomposition (VMD) and self-organizing map (SOM) neural network.

## 3. Based on the Kurtosis Criterion VMD Gear Fault Feature Extraction Method

### 3.1. VMD Signal Decomposition

The VMD algorithm was proposed by Dragomiretskiy and Zosso in 2014. The VMD algorithm decomposes the collected gear vibration signal $x(t)$ by constructing a variational model, and adaptively decomposes the gear vibration signal $x(t)$ by searching for the constrained variational optimal solution. The signal is adaptively decomposed into k IMFs $x_k(t)$ [23]:

Finally, the decomposed IMFs $x_k(t)$ are used to construct the squared *L*2 norm of the VMD algorithm is expressed as:

$$\min_{\{u_k\},\{\omega_k\}} \left\{ \sum_k \left\| \partial_t \left[ \left( \delta(t) + \frac{j}{\pi t} \right) \times x_k(t) \right] e^{-j\omega_k t} \right\|_2^2 \right\}$$
$$s.t. \sum_{k=1} u_k = x(t)$$
(1)

where: $\partial_t$ is the partial derivative of *t*, $x(t)$ is the original signal, $\omega_k$ is the bandwidth center frequency, and $\delta_t$ is the pulse signal. In this paper, $k = 4$ is derived based on the magnitude of the instantaneous frequency value.

The time-domain and frequency-domain diagrams of vibration signals collected under four working conditions of normal gear, gear with tooth wear, gear with tooth crack, and gear with tooth break are shown in Figure 4.

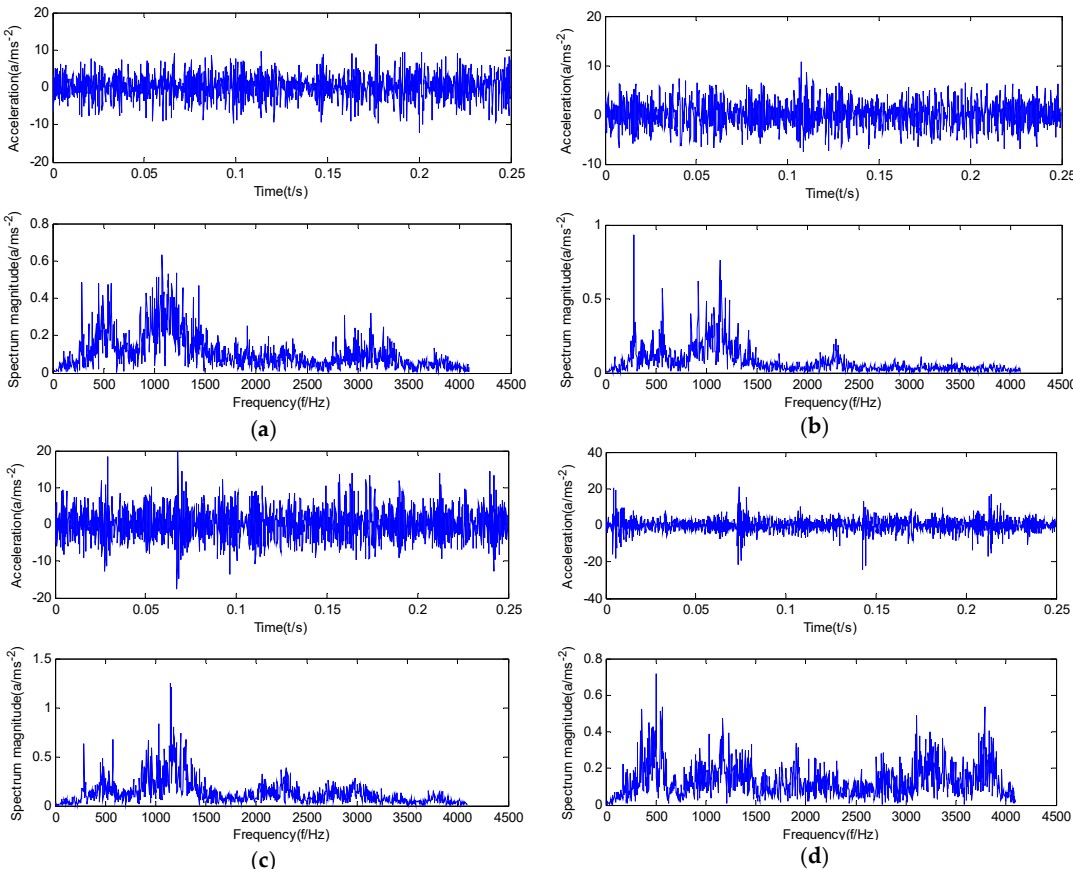

**Figure 4.** Time-domain and frequency-domain diagram of gear vibration signal: (**a**) Normal gear vibration signal, (**b**) gear with tooth wear vibration signal, (**c**) gear with tooth crack vibration signal, and (**d**) gear with tooth break vibration signal.

It can be seen from Figure 4 that the four types of gear failure signals are obtained after Fourier transform, and the four types of signals have the same frequency band peaks in the frequency spectrum. The noise contained in the signal and the acquisition time is also different. After performing the Fourier transform, the characteristic frequency of the signal may be masked. Therefore, a fault diagnosis algorithm is needed to intelligently diagnose gear faults.

In the VMD decomposition of the signal, the decomposition number $k$ and the penalty factor $\alpha$ are selected empirically. According to experience, the number of values of $k$ is often decomposed inaccurately, resulting in the value of $k$ is often too large. After VMD decomposition, some intermittent and useless frequency bands will appear. Therefore, the choice of $k$ value is also very important in VMD decomposition. In this paper, the instantaneous frequency mean is selected as the evaluation index, and the change of instantaneous frequency mean when the number of decomposition is 1–9 is selected as the basis for selecting the value of decomposition number $k$ [24]. Figure 5 respectively shows the instantaneous frequency mean size of the fault signal under the four working conditions of normal gear, gear with tooth wear, gear with tooth crack, and gear with tooth break at the decomposition number of 1–9:

According to the analysis of Figure 5, after the $k$ value is 5, the high frequency component is somewhat bent and intermittent. When the number of decomposition is greater than 4, some useless components may be decomposed. So $k = 4$.

The constrained variational problem in the formula is further solved, and the quadratic penalty factor $\alpha$ and *Lagrange* multiplication operator $\lambda(t)$ are introduced in consideration of the fact that the constraint becomes non-constraint. The quadratic penalty factor $\alpha$ guarantees the signal reconstruction accuracy in the noise environment, and the *Lagrange* multiplication operator $\lambda(t)$ keeps the constraint condition strict. The extended Lagrange is defined as Equation (2):

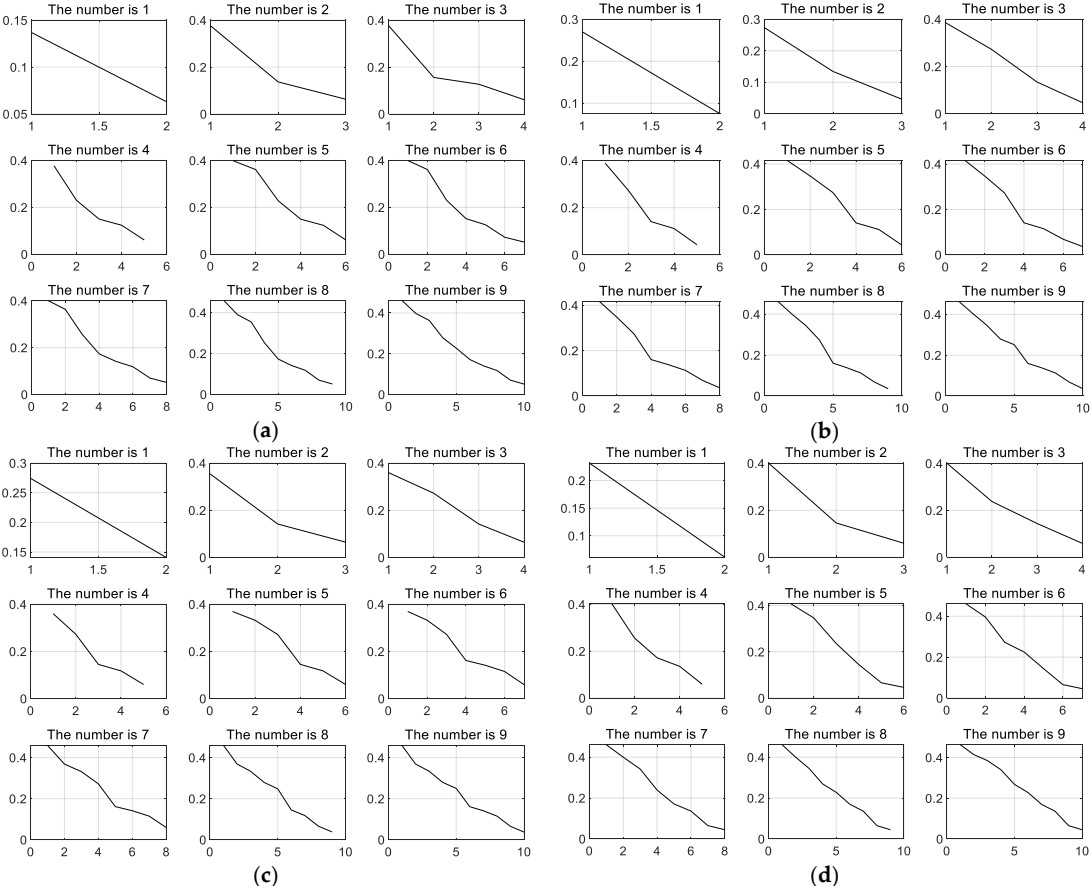

**Figure 5.** The instantaneous frequency mean of gear vibration signal at the decomposition number of 1–9: (**a**) Normal gear vibration signal, (**b**) gear with tooth wear vibration signal, (**c**) gear with tooth crack vibration signal, and (**d**) gear with tooth break vibration signal.

$$
\begin{aligned}
L(\{x_k\}, \{\omega_k\}, \{\lambda\}) = \\
\alpha \sum_k \left\| \partial_t \left[ \left( \delta(t) + \tfrac{j}{\pi t} \right) \times x_k(t) \right] e^{-j\omega_k t} \right\|_2^2 + \left\| f(t) - \sum_k x_k(t) \right\|_2^2 + \left\langle \lambda(t), f(t) - \sum_k x_k(t) \right\rangle
\end{aligned}
\tag{2}
$$

To solve the optimal solution of Equation (5), the alternating direction multiplication operator algorithm is used. The specific implementation steps of the algorithm are as follows:

Firstly, initialize the parameters, including: $\{\hat{x}_k{}^1\}, \{\hat{\omega}_k{}^1\}, \{\hat{\lambda}_k{}^1\}$, mode function is 4, error $\varepsilon$, output $\{x_k\}, \{\omega_k\}, \lambda$.

Step 1. Execution loop $n = n + 1$;

Step 2. Update $\{\hat{x}_k\}$ for all $\omega \geq 0$;

$$
\hat{x}_k{}^{n+1}(\omega) = \frac{\hat{f}(\omega) - \sum\limits_{i \neq k} \hat{x}_i(\omega) + \hat{\lambda}_i(\omega)/2}{1 + 2\alpha(\omega - \omega_k)^2}
\tag{3}
$$

$k \in [1, 4]$

Step 3. Update the modal center frequency $\{\hat{\omega}_k\}$;

$$\hat{\omega}_k{}^{n+1} = \frac{\int_0^\infty \omega |u(\omega)|^2 d\omega}{\int_0^\infty |u_k(\omega)|^2 d\omega}, k \in [1,4] \tag{4}$$

Step 4. Update $\lambda$;

$$\hat{\lambda}^{n+1}(\omega) = \hat{\lambda}^n(\omega) + \tau(\hat{f}(\omega) - \sum_k x_k{}^{n+1}(\omega)) \tag{5}$$

Step 5. Repeat Steps 1–4 until the iteration stop condition is satisfied;

$$\sum_k \left\| \hat{x}_k{}^{n+1} - \hat{x}_k{}^n \right\|_2^2 \Big/ \left\| \hat{x}_k{}^n \right\|_2^2 < \varepsilon \tag{6}$$

Step 6. At the end of iteration, four IMFs are obtained;

Where: $\hat{f}(\omega), \hat{x}_i(\omega), \hat{\lambda}(\omega)$ represents the Fourier transform of $f(\omega), x_i(\omega), \lambda(\omega)$, respectively, and $\varepsilon$ represents the discriminant accuracy.

After VMD decomposition of the signal, the penalty factor a was selected $\alpha$ is 2000 according to experience, and the discriminant accuracy $\varepsilon$ was. $10^{-7}$ After repeated experimental analysis, the *k* value was 4. In this paper, VMD decomposition time-domain waveform and frequency spectrum of vibration signals of normal gear, gear with tooth wear, gear with tooth crack, and gear with tooth break under four working conditions is shown in Figure 6.

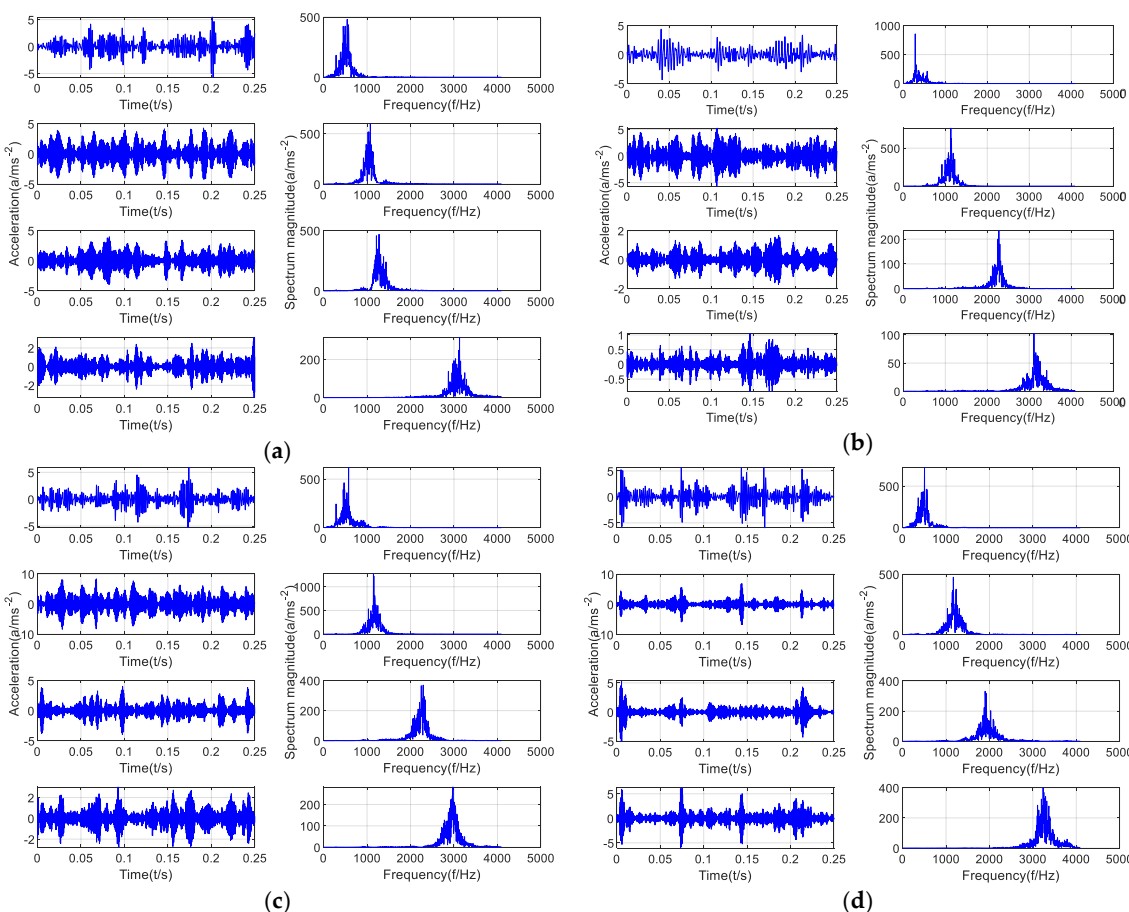

**Figure 6.** Gear vibration signal VMD decomposition time-domain waveform and spectrum: (**a**) Normal gear vibration signal, (**b**) gear with tooth wear vibration signal, (**c**) gear with tooth crack vibration signal, and (**d**) gear with tooth break vibration signal.

As can be seen from the above figure, the VMD concentrates the modal components near the center of each frequency, effectively improving the modemixing, thus verifying the correctness of the *k* selection.

After parameter optimization, the VMD algorithm is compared with EMD algorithm. In order to verify the effectiveness of the proposed method, a simulation signal is constructed to decompose EMD and VMD respectively, and the simulation signal [9] is as follows:

$$\begin{cases} x_1(t) = \cos(2\pi\omega_1 t) \\ x_2(t) = \frac{\cos(2\pi\omega_2 t)}{4} \\ x_3(t) = \frac{\cos(2\pi\omega_3 t)}{16} \\ n(t) = \text{Gaussian white noise} \\ x(t) = x_1(t) + x_2(t) + x_3(t) + n(t) \end{cases} \tag{7}$$

where, $\omega_1, \omega_2, \omega_3$ are the frequencies of each component signal, and their values are $\omega_1 = 3$, $\omega_2 = 25$, $\omega_3 = 289$, $t$ is time, $x_1(t), x_2(t), x_3(t)$ is fault signal, and $n(t)$ is Gaussian white noise. Figure 7 is the time-domain and frequency-domain diagram of the simulation signal $x(t)$.

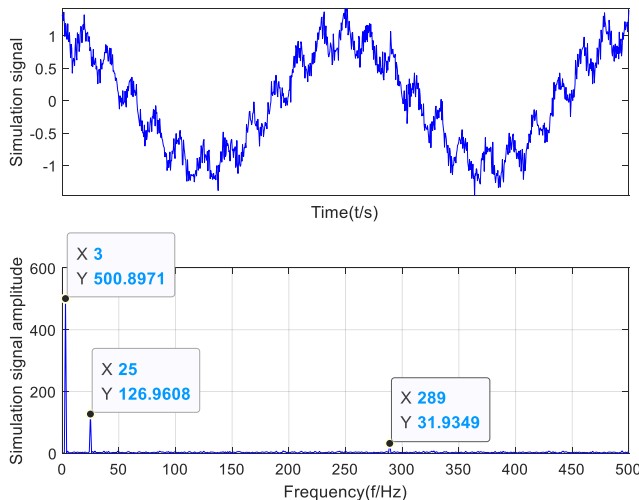

**Figure 7.** Time-domain and frequency-domain diagram of the simulation signal.

The simulation signals are decomposed by EMD and VMD, respectively, and the parameters in the VMD algorithm are determined by the parameters obtained after the above optimization. Where $k = 4$ and $\alpha = 2000$. The following Figures 8 and 9 respectively show the decomposition results of EMD and VMD.

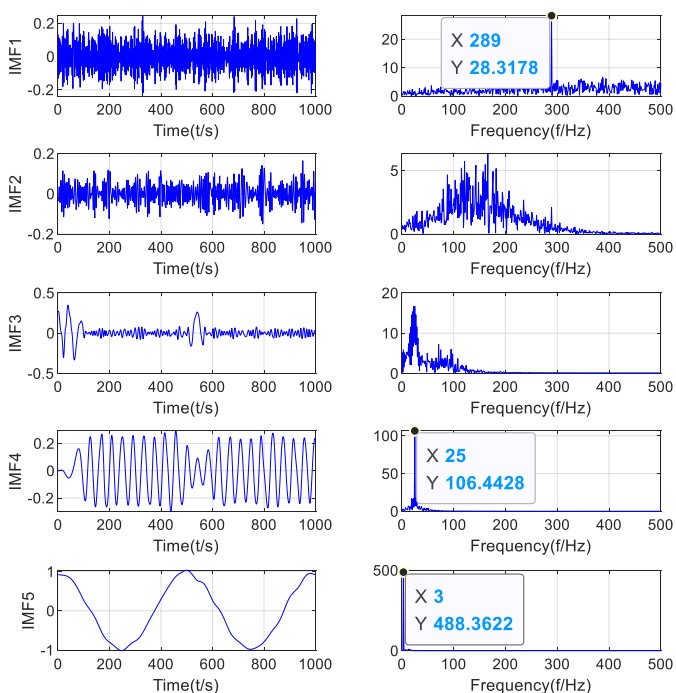

**Figure 8.** Empirical mode decomposition (EMD) decomposes the simulation signal.

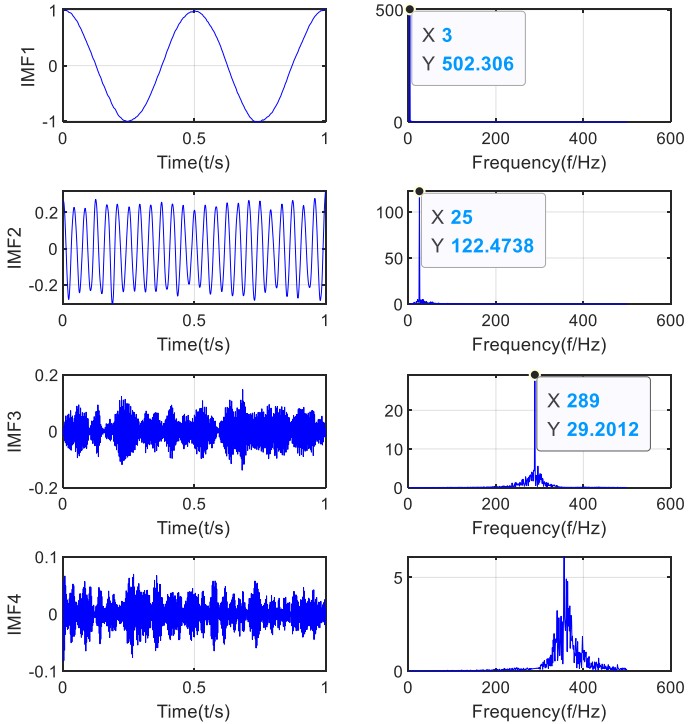

**Figure 9.** VMD decomposes the simulation signal.

It can be seen from Figure 8 that the frequency of the main fault signal can be obtained after EMD decomposition of the simulation signal. However, it can be seen from the frequency-domain diagram of IMF4 that it has mode mixing with IMF3 and IMF2, indicating that the component signal with characteristic frequency $\omega_2 = 25$ also exists in the component signals of IMF3 and IMF2. At the same time, the endpoint effect occurred at the left endpoint of the time-domain diagram of IMF4 in the EMD decomposition diagram, and its value was 0, which would have a bad effect on the Fourier transform of the component signal.

Then, the decomposition of gear fault signals collected by EMD under actual working conditions was compared with the above VMD decomposition effect to obtain the advantages of VMD in the decomposition process. Figures 10–13 show the decomposition of fault signals by EMD algorithm under normal gear, gear with tooth wear, gear with tooth crack, and gear with tooth break conditions.

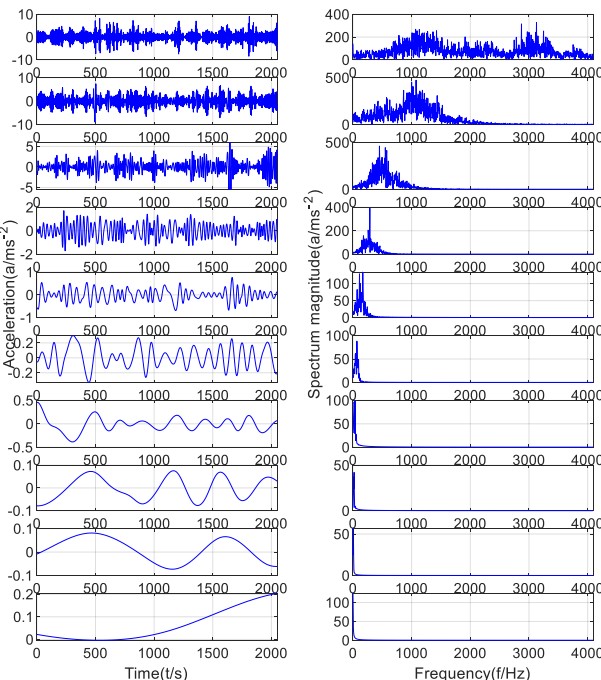

**Figure 10.** Normal gear fault signals decomposition by EMD algorithm.

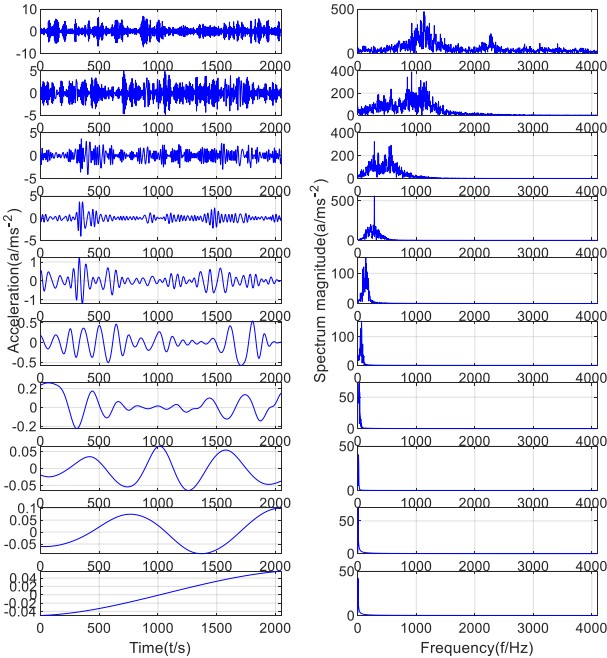

**Figure 11.** Gear with tooth wear fault signals decomposition by EMD algorithm.

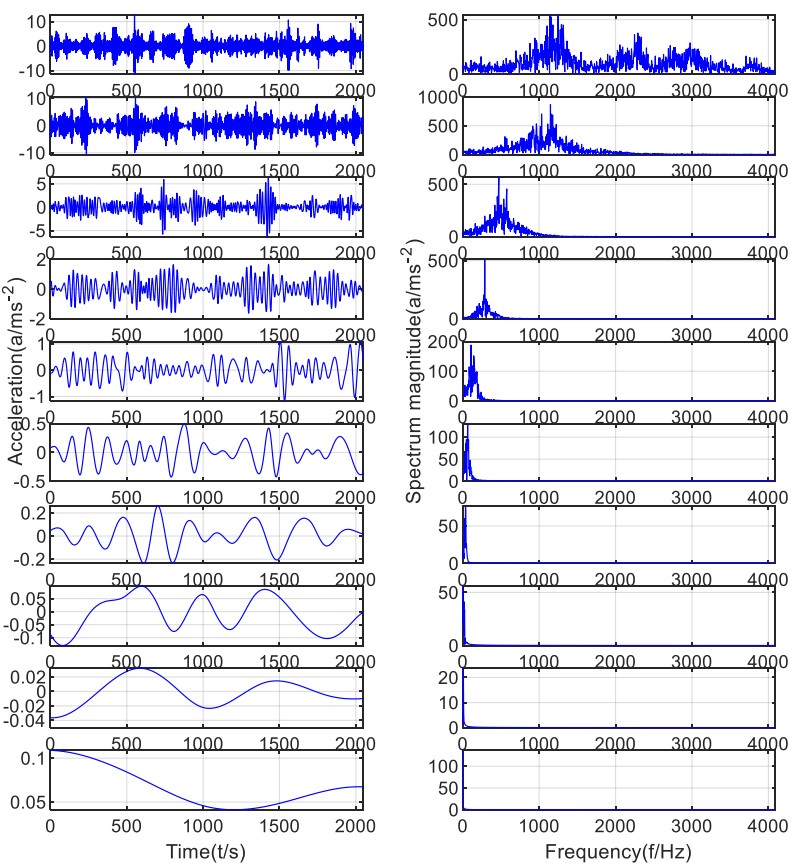

**Figure 12.** Gear with tooth crack fault signals decomposition by EMD algorithm.

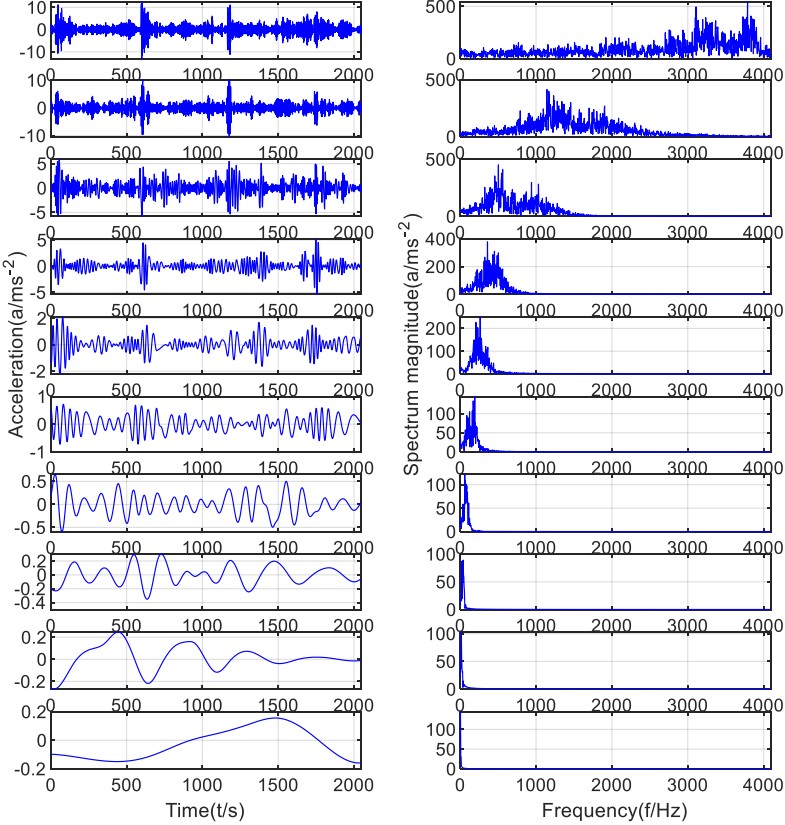

**Figure 13.** Gear with tooth break fault signals decomposition by EMD algorithm.

It can be seen from the results of the above gear fault signal after EMD decomposition and from the frequency-domain diagram of the IMFs that the phenomenon of mode mixing occurred in the frequency-domain, and finally the false component was decomposed, whose frequency value was almost close to 0, without any practical significance.From the results of VMD decomposition, it can be seen that this method can effectively avoid the phenomenon of mode mixing and endpoint effect in the EMD algorithm, and has certain help to improve the accuracy of gear fault diagnosis.

### 3.2. Feature extraction Based on Kurtosis Criterion VMD

Kurtosis is a dimensionless parameter, which reflects the numerical statistics of the random variable distribution characteristics of the signal, and also represents the normalized fourth order central moment of the signal. With the emergence and development of gear faults, the kurtosis of vibration signals increases. The higher the kurtosis value of the vibration signal, the more serious the gear fault and the easier of the gear faults information is to be extracted. Its mathematical formula is shown in Equation (8):

$$K = \frac{E(x - \mu)^4}{\sigma^4} \tag{8}$$

where, $\mu$ and $\sigma$ are the mean and standard deviation of the vibration signal, respectively.

Taking the instantaneous frequency mean as the VMD evaluation index, the number of VMD decomposition was determined after K=4, and the original vibration signal is decomposed into four components by VMD. Since the original vibration signal contains some characteristics of the original information after VMD decomposition, this paper extracts the kurtosis value of each mode to form feature vectors, and then normalizes the kurtosis value to form new feature vectors, which are input into the SOM neural network for gear fault mode classification and identification.

The 16 groups of data collected under 4 states were decomposed into VMD, the 4 IMF components decomposed by VMD were selected to extract kurtosis value, and then the feature vectors were formed. K1 represents the kurtosis of IMF1, K2 represents the kurtosis of IMF2, K3 represents the kurtosis of IMF3, and K4 represents the kurtosis of IMF4. The kurtosis value matrix of each component is shown in Table 1. After the kurtosis value of each component is extracted, in order to eliminate the influence of dimension, the kurtosis value matrix row vector normalization is adopted. The processing results are shown in Table 2.

**Table 1.** Kurtosis values of each intrinsic mode functions (IMFs).

| Gear Fault Type | Sample Number | K1 | K2 | K3 | K4 |
|---|---|---|---|---|---|
| Normal gear | 1 | 5.8307 | 1.3609 | 3.4593 | 2.6815 |
| | 2 | 5.8335 | 1.3690 | 3.4578 | 2.6883 |
| | 3 | 5.8331 | 1.3675 | 3.4592 | 2.6875 |
| | 4 | 5.8341 | 1.3603 | 3.4564 | 2.6883 |
| Gear with tooth wear | 1 | 6.7332 | 2.8831 | 4.9074 | 3.4868 |
| | 2 | 6.7742 | 2.8387 | 4.9679 | 3.4013 |
| | 3 | 6.7018 | 2.8033 | 4.9781 | 3.4829 |
| | 4 | 6.7793 | 2.8965 | 4.9508 | 3.4726 |
| Gear with tooth crack | 1 | 4.2817 | 1.9227 | 3.8432 | 2.7852 |
| | 2 | 4.2891 | 1.9228 | 3.8209 | 2.7972 |
| | 3 | 4.2999 | 1.9215 | 3.8397 | 2.7717 |
| | 4 | 4.2886 | 1.9225 | 3.8470 | 2.7886 |
| Gear with tooth break | 1 | 8.9575 | 7.7928 | 9.8567 | 4.6750 |
| | 2 | 8.9281 | 7.7706 | 9.8311 | 4.6933 |
| | 3 | 8.9978 | 7.7454 | 9.8819 | 4.6157 |
| | 4 | 8.9271 | 7.7580 | 9.8879 | 4.6864 |

**Table 2.** The kurtosis values of theIMFs are normalized.

| Gear Fault Type | Sample Number | K1 | K2 | K3 | K4 |
|---|---|---|---|---|---|
| Normal gear | 1 | 0.4373 | 0.1021 | 0.2595 | 0.2011 |
| | 2 | 0.4370 | 0.1026 | 0.2590 | 0.2014 |
| | 3 | 0.4370 | 0.1025 | 0.2592 | 0.2014 |
| | 4 | 0.4374 | 0.1020 | 0.2591 | 0.2015 |
| Gear with tooth wear | 5 | 0.3738 | 0.1601 | 0.2725 | 0.1936 |
| | 6 | 0.3767 | 0.1579 | 0.2763 | 0.1891 |
| | 7 | 0.3730 | 0.1560 | 0.2771 | 0.1939 |
| | 8 | 0.3746 | 0.1600 | 0.2735 | 0.1919 |
| Gear with tooth crack | 9 | 0.3337 | 0.1498 | 0.2995 | 0.2170 |
| | 10 | 0.3343 | 0.1499 | 0.2978 | 0.2180 |
| | 11 | 0.3351 | 0.1497 | 0.2992 | 0.2160 |
| | 12 | 0.3338 | 0.1496 | 0.2995 | 0.2171 |
| Gear with tooth break | 13 | 0.2863 | 0.2491 | 0.3151 | 0.1494 |
| | 14 | 0.2859 | 0.2489 | 0.3149 | 0.1503 |
| | 15 | 0.2880 | 0.2479 | 0.3163 | 0.1477 |
| | 16 | 0.2856 | 0.2482 | 0.3163 | 0.1499 |

## 4. SOM Neural Network Fault Diagnosis Model

### 4.1. SOM Neural Network Structure

The SOM neural network is used for regional division of data and regional classification of input variables, so as to study the distribution characteristics and topological relations of input variables. The principle is as follows: SOM neural network is an unsupervised clustering algorithm. Any data input to the SOM neural network forms a one-dimensional or two-dimensional planar topological structure array under its action, and then it is mapped into a one-dimensional or two-dimensional discrete graph. Finally, the result is output at the competitive layer, and the characteristic structure of the input mode remains unchanged. The input layer is used to receive the input mode and the contention layer is used to display the output mode. The neurons in the competition layer and the neurons in the input layer are fully connected, and the neurons in the competition layer are connected to each other to complete the function of pattern clustering.

SOM neural network is an unsupervised learning network that maps similar sample points in high-dimensional space to adjacent neurons in the network input layer. The nodes around the winning node respond to each other due to mutual influence. Therefore, the weight vectors connected by the winning node and all nodes in the winning field adjust to the input direction, gradually reducing the distance between them. Then through self-organizing learning and repeated learning. The spatial distribution density of the connection weights is consistent with the probability distribution of the input mode, and finally the feature map is formed at the input layer. The structure diagram of the SOM neural network is shown in Figure 14.

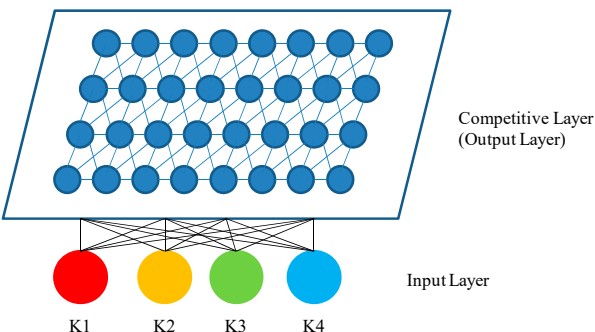

**Figure 14.** SOM neural network structure.

### 4.2. SOM Neural Network Algorithm

(1) Weight initialization
A random value within the interval of $[0,1]$ is given to the initial weight vector $w_{ij}(0)$.
(2) Sampling
Take a random sample from the input vector with a certain probability.
(3) Determining winning neurons
Within the set step size N of SOM neural network algorithm, the following criteria are used to determine the winning neuron:

$$i(x) = \underset{i}{\mathrm{argmax}} \; y_i(x) \tag{9}$$

where: $y_i(x)$ is the steady state output of the neural network without feedback:

$$y_i(x) = \sum_{j=1}^{n} w_{ij} x_j \tag{10}$$

A topological field of winning neuron C is defined as $N_c$, then:

$$\begin{cases} y_i = 1, i \in N_c \\ y_i = 0, i \notin N_c \end{cases}$$

Otherwise:

$$w_{ij}(n+1) = w_{ij}(n)$$

$0 < \alpha(n) < 1$ is the learning factor, and step (2) is returned after weight training until $N_c$ or $\alpha(n)$ meets the requirements.

(4) Enter the next vector until all the samples are learned. Training samples were input into SOM neural network for training, and a standard SOM neural network was obtained. Then, input the test samples data into the SOM neural network. If the position of the output neuron in the output layer is the same as that of a sample data, the sample to be tested is of the corresponding type.

## 5. Fault Diagnosis and Result Analysis of SOM Neural Network

The standard component kurtosis values are selected for training. The sample number 1 and sample number 2 of the normal gear data set, the sample number 5 and the sample number 6 of the gear with tooth wear data set, the sample number 9 and the sample number 10 of the gear with tooth crack data set, and the sample number 13 and sample number 14 of the gear with tooth break data set are selected to form a training sample matrix input into the SOM neural network for training. After the training of SOM neural network, the remaining 8 sets of samples were input into the standard SOM neural network for gear fault diagnosis as the unknown gear fault test sample matrix.

In this paper, $16 \times 4$ arranged neurons are adopted, whose topological structure is hexagonal. The structure of the composed SOM neural network is shown in Figure 15.

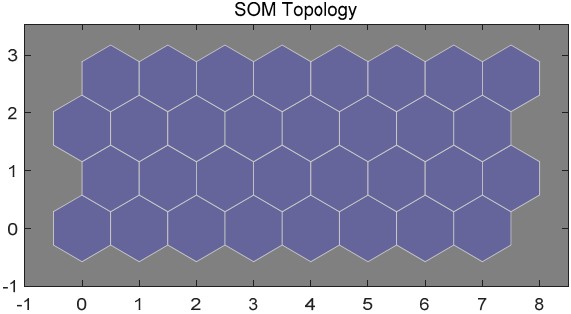

**Figure 15.** SOM neuron structure.

The SOM neural network neighbor neuron connections as shown in Figure 16. There are 32 neurons in total.Generally, the sequence number of neurons in the lower left corner of SOM neural network topological structure graph is marked as 1, and the sequence number increases successively from left to right and from bottom to top. The 32nd neuron is located in the upper right corner of Figure 16.

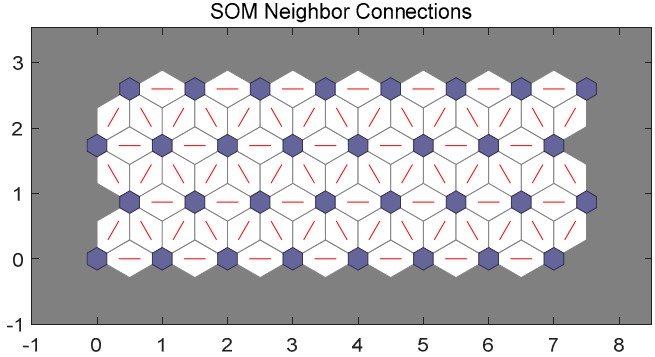

**Figure 16.** SOM neural network neighbor neuron connections.

*Fault Diagnosis and Result Analysis*

Eight sets of data were extracted from 16 sets of sample data, including one set for each gear states. The remaining 8 sets of data were tested as test data sets. The kurtosis value normalization matrix was input into SOM neural network for classification and identification, and the following results were the training times of 100.

The kurtosis value normalization matrix in Table 2 is used as training data to train the SOM neural network.As the number of training steps increases, theweight vector is constantly adjusted and the magnitude of theweight vector is also changed.Where, the weight connection between the input sample and the competition layer neuronsare shown in Figure 17, where the lightest color hexagon weight value is the smallest, and the black hexagon weight value is usually 0.

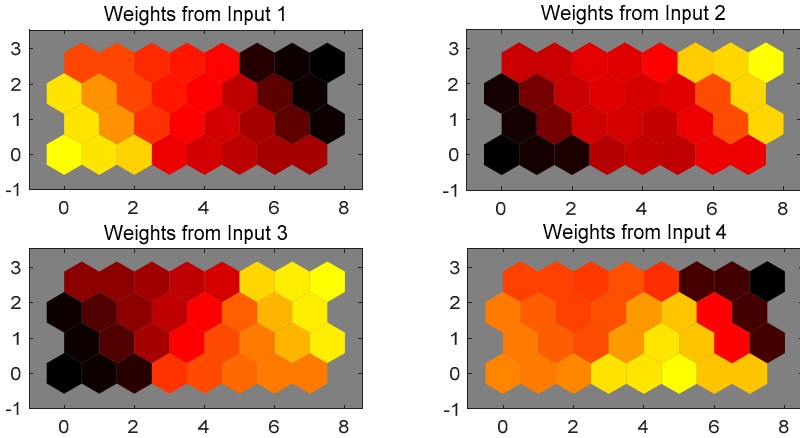

**Figure 17.** Weight connection of SOM neural network neuronswhen the number of training is 100.

As shown in Figure 18, the hexagon represents a neuron on the competition layer, the node of the neuron is represented by 32 equal-sized hexagons, and the short line between the hexagons represents the interconnection between the neurons.The difference in color between the diamond blocks located around the neurons indicates the difference in distance between the neurons. The SOM neural network forms clusters based on the distance between the neurons, and there is no clear boundary between the clustering results. From the color (light to dark) of the diamond block, the darker the color that the block is, the farther the distance between the neurons.

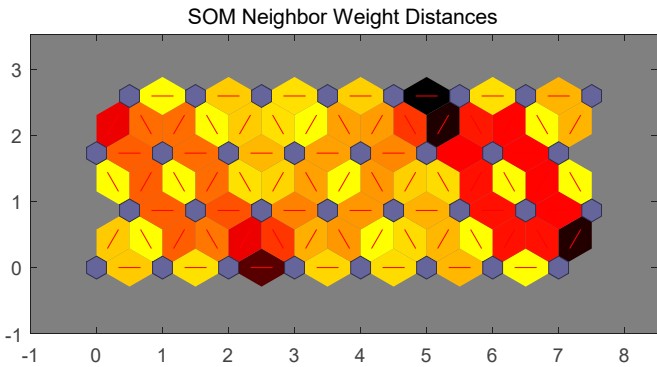

**Figure 18.** Distance distribution of SOM neural network neurons when the number of training is 100.

Figure 19 is the recognition result of training times of 100:

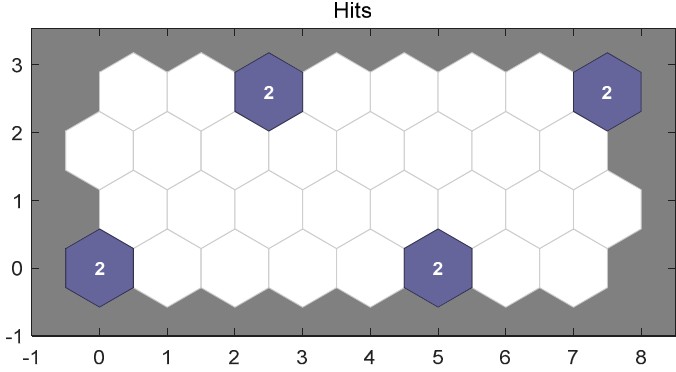

**Figure 19.** SOM neural network classification results of gear fault when the number of training is 100.

According to the clustering results of SOM neural network, adjacent neurons can be regarded as the same species, while neurons with more distant neighbors can be regarded as different species. In Table 3, the numerical values of the classification results indicate the location number of SOM neural network neurons, and the adjacent number indicates the adjacent neurons, which are regarded as the same species. As shown in Figure 12, the four fault types of gear fault, normal gear, gear with tooth wear, gear with tooth crack, and gear with tooth break are effectively distinguished. The accuracy of gear fault of SOM neural network is 100% when the number of trainings is 100. Among them, Table 3 shows the classification results of the training samples numbered 1, 2, 5, 6, 9, 10, 13, and 14, which were input into the SOM neural network as the training sample matrix and the training times were 100.

**Table 3.** Classification results for 100 training sessions.

| Gear Fault Type | Training Steps | Normal Gear | | Gear with Tooth Wear | | Gear with Tooth Crack | | Gear with Tooth Break | |
|---|---|---|---|---|---|---|---|---|---|
| Sample number | 100 | 1 | 2 | 5 | 6 | 9 | 10 | 13 | 14 |
| Classification results | | 1 | 1 | 6 | 6 | 20 | 20 | 32 | 25 |

Table 4 shows the classification results of the samples numbered 3, 4, 7, 8, 11, 12, 15, and 16, which were input into the SOM neural network as the test samples matrices after 100 times of training. By comparing the classification results in Table 3, it can be seen that all test samples have been correctly classified into the corresponding fault categories, and the fault diagnosis recognition rate is 100%.

**Table 4.** The classification results of the test samples at 100 training times.

| Gear Fault Type | Training Steps | Normal Gear | | Gear with Tooth Wear | | Gear with Tooth Crack | | Gear with Tooth Break | |
|---|---|---|---|---|---|---|---|---|---|
| Sample number | | 3 | 4 | 7 | 8 | 11 | 12 | 15 | 16 |
| Classification results | 100 | 1 | 1 | 6 | 6 | 20 | 20 | 32 | 25 |

Figures 17–19 show that when the training times are 100 times, 8 test samples of SOM neural network fault diagnosis model classifier correspond with the training samples of standard SOM neural network, and its fault diagnosis accuracy is up to 100%. It effectively proves the effectiveness of SOM neural network in the classification and identification of gear fault diagnosis.

In order to obtain the identification effect advantage of the SOM neural network when the training times are 100 times and the training times are 10, 50, 500, and 1000 times, the gear failure accuracy is compared, respectively, in Figures 20–23.

Figure 20 shows the recognition result of the training number 10:

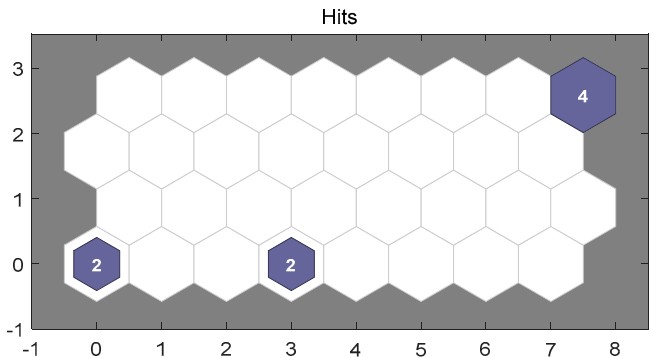

**Figure 20.** SOM neural network classification results of gear fault when the number of training is 10.

As can be seen from the identification results in Figure 20, the four fault types of gear, normal gear, gear with tooth wear, gear with tooth crack, and gear with tooth break were classified into three fault states and onegear fault state was not identified. From the identification results in Tables 5 and 6, it can be concluded that normal gear and gear with tooth wear are identified into the same category. This is the classification result of SOM neural network with 10 times of training, and Table 5 is the classification result of SOM neural network with 10 times of training as the training sample matrix with sample numbers 1, 2, 5, 6, 9, 10, 13, and 14.

**Table 5.** Classification results for 10 training sessions.

| Gear Fault Type | Training Steps | Normal Gear | | Gear with Tooth Wear | | Gear with Tooth Crack | | Gear with Tooth Break | |
|---|---|---|---|---|---|---|---|---|---|
| Sample number | | 1 | 2 | 5 | 6 | 9 | 10 | 13 | 14 |
| Classification results | 10 | 32 | 32 | 32 | 32 | 4 | 4 | 1 | 1 |

Table 6 shows the classification results of the sample numbered 3, 4, 7, 8, 11, 12, 15, and 16, which were input into the SOM neural network as the test sample matrices after 10 training times. By comparing the classification results in Table 5, it can be seen that the classification results of the test samples from the fault training samples are not good, in which there is an identification error in the fault state of gear with tooth wear and gear with tooth break respectively, resulting in a fault diagnosis identification rate of 75%.

**Table 6.** The classification results of the test samples at 10 training times.

| Gear Fault Type | Training Steps | Normal Gear | | Gear with Tooth Wear | | Gear with Tooth Crack | | Gear with Tooth Break | |
|---|---|---|---|---|---|---|---|---|---|
| Sample number | | 3 | 4 | 7 | 8 | 11 | 12 | 15 | 16 |
| Classification results | 10 | 32 | 32 | 32 | 32 | 4 | 4 | 1 | 1 |

Figure 21 shows the recognition result of the training number 50:

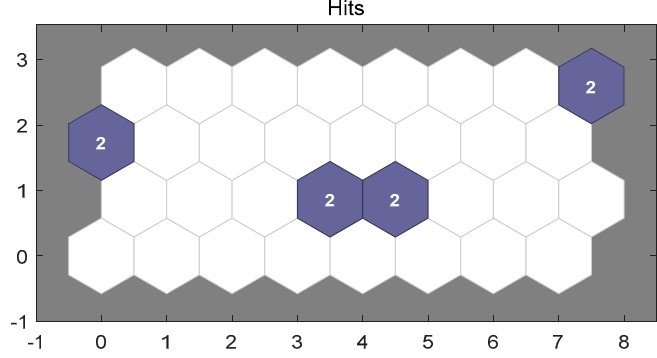

**Figure 21.** SOM neural network classification results of gear fault when the number of training is 50.

As can be seen from the identification results in Figure 21, the four fault types of gear, normal gear, gear with tooth wear, gear with tooth crack, and gear with tooth break, were classified into three fault states, and also onegear fault state was not identified. From the identification results in Tables 7 and 8, it can be concluded that gear with tooth wear and gear with tooth crack are identified into the same category. This is the classification result of SOM neural network with 50 times of training, and Table 7 is the classification result of SOM neural network with 50 times of training as the training sample matrix with sample numbers 1, 2, 5, 6, 9, 10, 13, and 14.

**Table 7.** Classification results for 50 training sessions.

| Gear Fault Type | Training Steps | Normal Gear | | Gear with Tooth Wear | | Gear with Tooth Crack | | Gear with Tooth Break | |
|---|---|---|---|---|---|---|---|---|---|
| Sample number | | 1 | 2 | 5 | 6 | 9 | 10 | 13 | 14 |
| Classification results | 50 | 17 | 17 | 13 | 13 | 12 | 12 | 32 | 32 |

Table 8 shows the classification results of the sample numbered 3, 4, 7, 8, 11, 12, 15, and 16, which were input into the SOM neural network as the test sample matrices after 50 training times. By comparing the classification results in Table 7, it can be seen that the classification results of the test samples from the fault training samples are not good, in which there is an identification error in the fault state of gear with tooth wear and gear with tooth break, respectively, resulting in a fault diagnosis identification rate of 75%.

**Table 8.** The classification results of the test samples at 50 training times.

| Gear Fault Type | Training Steps | Normal Gear | | Gear with Tooth Wear | | Gear with Tooth Crack | | Gear with Tooth Break | |
|---|---|---|---|---|---|---|---|---|---|
| Sample number | | 3 | 4 | 7 | 8 | 11 | 12 | 15 | 16 |
| Classification results | 50 | 17 | 17 | 13 | 13 | 12 | 12 | 32 | 32 |

Figure 22 shows the recognition result of the training number 500:

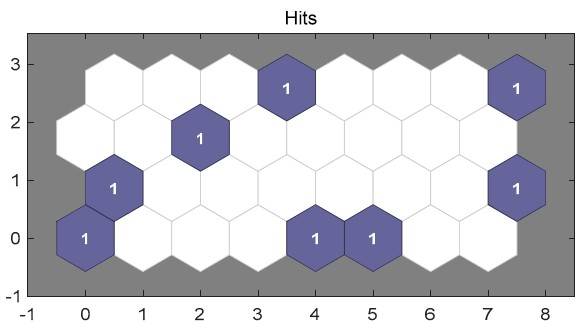

**Figure 22.** SOM neural network classification results of gear fault when the number of training is 500.

As can be seen from the identification results in Figure 22, the four fault types of gear, normal gear, gear with tooth wear, gear with tooth crack, and gear with tooth break were classified into six fault states and two unknown fault states emerged. This is the classification result of SOM neural network with 500 times of training, and Table 9 is the classification result of SOM neural network with 500 times of training as the training sample matrix with sample numbers 1, 2, 5, 6, 9, 10, 13, and 14.

Table 10 shows the classification results of the sample numbered 3, 4, 7, 8, 11, 12, 15, and 16, which were input into the SOM neural network as the test sample matrices after 500 training times. By comparing the classification results in Table 9, it can be seen that the classification results of the test samples from the fault training samples are not good, in which there is an identification error in the fault state of gear with tooth wear and gear with tooth break respectively, resulting in a fault diagnosis identification rate of 75%.

**Table 9.** Classification results for 500 training sessions.

| Gear Fault Type | Training Steps | Normal Gear | | Gear with Tooth Wear | | Gear with Tooth Crack | | Gear with Tooth Break | |
|---|---|---|---|---|---|---|---|---|---|
| Sample number | | 1 | 2 | 5 | 6 | 9 | 10 | 13 | 14 |
| Classification results | 500 | 1 | 9 | 28 | 19 | 5 | 6 | 32 | 16 |

**Table 10.** The classification results of the test samples at 500 training times.

| Gear Fault Type | Training Steps | Normal Gear | | Gear with Tooth Wear | | Gear with Tooth Crack | | Gear with Tooth Break | |
|---|---|---|---|---|---|---|---|---|---|
| Sample number | | 3 | 4 | 7 | 8 | 11 | 12 | 15 | 16 |
| Classification results | 500 | 9 | 1 | 11 | 27 | 4 | 5 | 32 | 24 |

Figure 23 shows the recognition result of the training number 1000:

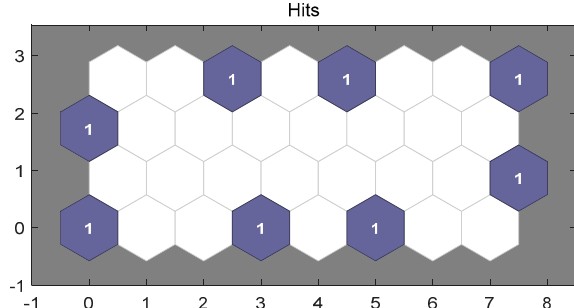

**Figure 23.** SOM neural network classification results of gear fault when the number of training is 1000.

As shown in Figure 23, the four fault types of gear fault, normal gear, gear with tooth wear, gear with tooth crack, and gear with tooth break were classified into eight fault states and four unknown fault states emerged. When the number of training sessions of SOM neural network is 1000, Table 11 shows the classification results of the training samples numbered 1, 2, 5, 6, 9, 10, 13, and 14 as training samples matrix input to SOM neural network when the number of training sessions is 1000.

**Table 11.** Classification results for 1000 training sessions.

| Gear Fault Type | Training Steps | Normal Gear | | Gear with Tooth Wear | | Gear with Tooth Crack | | Gear with Tooth Break | |
|---|---|---|---|---|---|---|---|---|---|
| Sample number | | 1 | 2 | 5 | 6 | 9 | 10 | 13 | 14 |
| Classification results | 1000 | 1 | 17 | 28 | 11 | 14 | 5 | 32 | 16 |

Table 12 shows the classification results of samples numbered 3, 4, 7, 8, 11, 12, 15, and 16, which were input into the SOM neural network as the test sample matrices after 1000 times of training. By comparing the classification results in Table 11, it can be seen that the classification results of test samples from fault training samples are not good, in which there is one identification error in the fault state of gear with tooth crack and gear with tooth break respectively, and two identification errors in the fault state of gear with tooth wear, resulting in a fault diagnosis identification rate of 50%.

**Table 12.** The classification results of the test samples at 1000 training times.

| Gear Fault Type | Training Steps | Normal Gear | | Gear with Tooth Wear | | Gear with Tooth Crack | | Gear with Tooth Break | |
|---|---|---|---|---|---|---|---|---|---|
| Sample number | | 3 | 4 | 7 | 8 | 11 | 12 | 15 | 16 |
| Classification results | 1000 | 17 | 2 | 20 | 20 | 14 | 14 | 32 | 24 |

Figures 19–23 shows that when the training times are 100 times, 8 test samples of SOM neural network fault diagnosis model classifier correspond with the training samples of standard SOM neural network, and its fault diagnosis accuracy is up to 100%. It effectively proves the effectiveness of SOM neural network in the classification and identification of gear fault diagnosis. However, when the training times were 10 times and 50 times, the 8 test samples of SOM neural network model classifier showed onegear fault state was not identified. And when the training times were 500 times and 1000 times, the 8 test samples of SOM neural network model classifier showed unrecognized types. This is not an identification error, which means that as the number of training sessions increases it is the same for each sample, but it is not divided into categories, and there is an unrecognizable type. At this time, if you increase the number of training, there is no practical significance.

From the comparison of the accuracy of gear fault recognition of the SOM neural network in different training times in Table 13, it can be obtained that the gear fault recognition accuracy of the SOM neural network is the highest when the training times are 100 times. So the training times of the SOM neural network are 100times.

**Table 13.** The accuracy of SOM neural network for gear fault identification under different training times.

| Training Steps | Fault Diagnosis Identification Rate |
|---|---|
| 10 | 75% |
| 50 | 75% |
| 100 | 100% |
| 500 | 75% |
| 1000 | 50% |

From the above results, it can be concluded that the effect of SOM neural network is the best when the training times are 100times. The EMD algorithm mentioned in the article will have adverse effects such as mode mixing in the gear fault signal decomposition. Here, the EMD algorithm is combined with the SOM neural network to identify the gear fault states, and then compared with the VMD algorithm and the SOM neural network recognition results.

First, the gear fault signal was decomposed by EMD algorithm, then the kurtosis value was extracted, and then it was normalized as the input vector of SOM neural network. According to the above results, the gear fault identification rate of SOM neural network was the highest when the training times were 100times.Therefore, the training times of SOM neural network are 100 times. The following Figure 24 shows the recognition diagram of gear fault signal by EMD algorithm is combined with the SOM neural network with 100 training times.

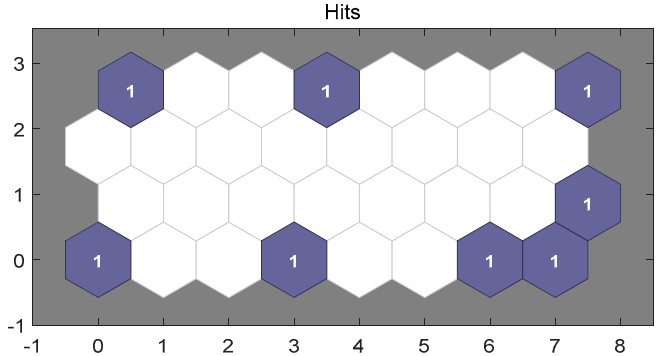

**Figure 24.** SOM neural network classification results of gear fault when the number of training is 100 by EMD.

As shown in Figure 24, the four fault types of gear fault, normal gear, gear with tooth wear, gear with tooth crack, and gear with tooth break were classified into six fault states and two unknown fault states emerged. When the number of training sessions of SOM neural network is 100, Table 14 shows the classification results of the training samples numbered 1, 2, 5, 6, 9, 10, 13, and 14 as training samples matrix input to SOM neural network when the number of training sessions is 100.

**Table 14.** Classification results for 1000 training sessions by EMD.

| Gear Fault Type | Training Steps | Normal Gear | | Gear with Tooth Wear | | Gear with Tooth Crack | | Gear with Tooth Break | |
|---|---|---|---|---|---|---|---|---|---|
| Sample number | | 1 | 2 | 5 | 6 | 9 | 10 | 13 | 14 |
| Classification results | 100 | 32 | 7 | 25 | 28 | 8 | 16 | 4 | 1 |

Table 15 shows the classification results of samples numbered 3, 4, 7, 8, 11, 12, 15, and 16, which were input into the SOM neural network as the test sample matrices after 100 times of training. By comparing the classification results in Table 14, it can be seen that the classification results of test samples from fault training samples are not good, in which there is one identification error in the fault state of gear with tooth crack and gear with tooth break respectively, and two identification errors in the fault state of gear with tooth wear, resulting in a fault diagnosis identification rate of 50%.

As can be seen from the comparison results in Tables 14 and 15, the EMD algorithm will not only generate mode mixing in the process of gear signal decomposition, but also reduce the accuracy of gear fault identification.

**Table 15.** The classification results of the test samples at 1000 training times by EMD.

| Gear Fault Type | Training Steps | Normal Gear | | Gear with Tooth Wear | | Gear with Tooth Crack | | Gear with Tooth Break | |
|---|---|---|---|---|---|---|---|---|---|
| Sample number | | 3 | 4 | 7 | 8 | 11 | 12 | 15 | 16 |
| Classification results | 100 | 7 | 22 | 26 | 27 | 8 | 32 | 6 | 7 |

According to the comparison results of Tables 3 and 4 with Tables 14 and 15, the advantages of VMD algorithm over EMD algorithm can be obtained, and the optimal training times of SOM can also be obtained. However, there are many classification algorithms. So how do we evaluate the advantages of SOM neural network over other classification algorithms? Here, the SVM algorithm is used for gear fault classification instead of SOM neural network, and the calculation results are shown in Table 16.

**Table 16.** The accuracy of gear fault diagnosis by different decomposition algorithms and classification algorithms.

| Decomposition Algorithm | Classification Algorithm | Fault Diagnosis Identification Rate |
|---|---|---|
| VMD | SOM | 100% |
| VMD | SVM | 25% |
| EMD | SOM | 75% |
| EMD | SVM | 14.29% |

According to the comparison of fault diagnosis accuracy in Table 16, compared with the EMD algorithm, the VMD algorithm can avoid mode mixing and other shortcomings in the EMD algorithm in terms of decomposition effect. In terms of the accuracy of gear fault diagnosis, the VMD algorithm has a higher accuracy than the EMD algorithm. As for the classification algorithm recognition rate of the SOM neural network, the overall fault diagnosis accuracy of the SOM neural network is much higher than that of the SVM algorithm whether it is matched with the EMD algorithm or the VMD algorithm.Therefore, the algorithm of VMD combined with SOM neural network has better advantages and better application prospects in gear fault diagnosis.

## 6. Conclusions

In this paper, a gear fault diagnosis method based on kurtosis criterion VMD and SOM neural network is proposed, and the following conclusions are obtained:

(1) The VMD is selected to decompose the gear acceleration signal, which can effectively improve the mode mixing phenomenon of EMD, LMD, and other decomposition methods. Then, the kurtosis normalized value of components is selected as the feature vectors, which can not only retain the effective fault information in the signal, but also effectively eliminate noise interference.

(2) The instantaneous frequency mean is selected as the basis of VMD decomposition number, which can effectively select the decomposition number in the VMD process. Through the above process, more representative components can be selected after VMD decomposition.

(3) The SOM neural network has the ability of dimensionality reduction of multidimensional spatial data, which can be trained with fewer samples to obtain the spatial topological relationship of classified data and obtain higher accuracy. At the same time, SOM neural network can also conduct fewer training steps and obtain higher accuracy.

**Author Contributions:** Conceptualization, D.X. and J.D.; methodology, J.D.; writing—original draft preparation, D.X. and J.D.; writing—review and editing, J.D., D.X., X.L., and L.H.; funding acquisition, D.X. and X.L.

**Funding:** This research was supported by National Natural Science Foundation of China (No. 51875195, 51875196).

**Conflicts of Interest:** The authors declare no conflict of interest.

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
