# Peer review of "Gear Fault Diagnosis Based on Kurtosis Criterion VMD and SOM Neural Network"

_applsci, doi:10.3390/app9245424_

Round 1
Reviewer 1 Report
The paper presents a gear fault diagnosis method based on kurtosis criterion variational mode decomposition (VMD) and self-organizing map (SOM) neural network. In overall, there is a slight novelity of VMD plus SOM. However, big improvements are required. First, More relevant literature review should be involved, e.g.Structural Health Monitoring Framework Based on Internet of Things: A Survey, IEEE INTERNET OF THINGS JOURNAL, vol. 4, no. 3, pp. 619-635, Feb 2017. Multidimensional Tensor-Based Inductive Thermography With Multiple Physical Fields for Offshore Wind Turbine Gear Inspection, IEEE TRANSACTIONS ON INDUSTRIAL ELECTRONICS, vol. 63, no. 10, pp. 6305-6315, Sep 2016. Variational Mode Decomposition Linked Wavelet Method for EMAT Denoise with Large Lift-off Effect, NDT and E International, vol. 107, pp. 1-15, Jul 2019. In addition, the detail explain why VMD + SOM is the best team for analuzing the data. Each of EMD or SOM should be shorten the contents as they are well known techniques. Moreover, the more comparison works should involved to prove why this one is better than other techniques.Author Response
Please see the attachment.

Reviewer 2 Report
I found the paper very interesting and well written, it has been easy to follow the work. The use of English is good; I haven’t found any typo nor mistake. Both the abstract and the introduction are clear and describe the aim of the work and the basis of the research. The bibliographic references are numerous, up to date and well connected with the text, making it easy to find additional information. The new solution presented as the core of the work, that is the use of VMD and SOM as diagnosis method, is quite interesting and, according to the results, looks very promising. I therefore recommend its publication.
Reviewer 3 Report
The paper deals with the gear diagnosis for several types of gear defects. The feature extraction is based on kurtosis criterion variational mode decomposition (VMD) and the classification system is a self-organizing map (SOM) neural network.
In the introduction, it is said that the wavelet analysis gives poor signal decomposition, since the wavelet basis function can not be changed during the process. This is a hard affirmation, considering that the wavelet analysis has been widely accepted and used in diagnosis of rotating elements in the literature with very good results, and it has proven its effectiveness in this area, but of course the wavelet function must be properly selected. Reviewer has found several mistakes in the text, like missing spaces after dots, wrong or not defined acronyms like EEMD, etc. In figure 2 images should be larger. The defects are not properly seen, but they seem to be very big. What do authors mean when they say that the number of trainings is 100? What is the difference between one training and another? Why 100? This should be properly clarified in the text. In reviewer's opinion, the data sets for each defect condition can be clearly distinguished even from the temporary signals and FFT's (figure 4). This makes difficult to appreciate the advantages of the VMD in this case with respect to the EMD, for example. It would be very interesting to see the comparison between the results obtained using the VMD and the EMD, since the authors affirm that the VMD technique is used to avoid to the endpoint effect and mode confusion in the EMD technique.Author Response
Please see the attachment.

Round 2
Reviewer 1 Report
The revised paper is ok for accept
Reviewer 3 Report
Thank you for your effort.